# Dynamics of Team Learning Behaviours: The Effect of Time and Team Culture

**DOI:** 10.3390/bs12110449

**Published:** 2022-11-14

**Authors:** Margarida Pinheiro, Teresa Rebelo, Paulo Renato Lourenço, Bruno de Sousa, Isabel Dimas

**Affiliations:** 1Faculty of Psychology and Educational Sciences, University of Coimbra, 3000-115 Coimbra, Portugal; 2Centre for Business and Economics Research (CeBER), Faculty of Economics, University of Coimbra, 3004-512 Coimbra, Portugal; 3Center for Research in Neuropsychology and Cognitive and Behavioral Intervention, University of Coimbra, 3000-115 Coimbra, Portugal

**Keywords:** team learning behaviours, team culture, temporal dynamics

## Abstract

This research study focused on team learning behaviours, particularly the extent to which teams use learning behaviours over time, as well as the influence of different team cultures on learning behaviours over time. Data from 33 university project teams were collected longitudinally at three moments (beginning, halfway point, and end of the project) and the analysis was conducted through growth modelling. A linear relationship between time and team learning through experimenting behaviour was found, suggesting that experimenting behaviour tends to increase over time in project teams. Moreover, the early development of team cultures that promote mutual understanding and good interpersonal relationships, the accomplishment of objectives, flexibility, and the search for alternative ways to perform tasks/problem solving are conducive to experimenting behaviours from the beginning of the teamwork. This study highlights the relevance of the temporal dynamics of team learning behaviours and their interaction with team culture.

## 1. Introduction

Research on team processes and outcomes began around the end of the 1980s and has continued to grow, both in volume and richness of the themes explored, e.g., [1,2,3,4]. In a working environment which is more dynamic and fluid than ever, with looser boundaries, more heterogeneity, competing demands, and more technology [3], teams have the potential to form flexible and creative working units, organized in such a way as to perform complex and dynamic tasks. Consequently, they have become the basic building blocks of modern organizational designs [5,6]. According to [1], a team is a “collection of individuals who are interdependent in their tasks, who share responsibility for outcomes, who see themselves and who are seen by others as an intact social entity embedded in one or more larger social systems and who manage their relationships across organizational boundaries”. The shared responsibility for outcomes and the necessary interdependence required of members to achieve team goals leads to the recognition of learning as a critical team process [7]. 

Considering that the ability of an organization to learn more quickly than others can be viewed as one of the best sources of a competitive advantage and that learning in organizations occurs in individuals and teams [8], team learning is a relevant process for current organizations. It can be conceptualized as an outcome variable of the team, e.g., [9], but also as a process, stressing the importance of team learning behaviours (e.g., [10,11,12,13]) and its significant role in helping teams and organizations to adjust to their environment (e.g., [14,15]). Following this latest stream of research, team learning can be considered as the behavioural patterns in which a team takes action, obtains and reflects on feedback, and makes changes to adapt or improve [10]. 

In the last years, the identification of the conditions that foster team learning has received increasing interest from the scientific community [10,12,14,16,17]. Among these conditions, team culture, i.e., team members’ beliefs about “the way things are done” in the team [18] has been identified as a potential factor to be considered in understanding team learning [19,20]. In fact, teams are integrated into an organizational context, and the team itself provides an environment to its members [21]. Team culture is defined as an emergent and simplified set of rules and actions, work capability expectation, and perceptions that are shared by team members, developed by themselves, and enacted after a range of team members’ interactions [22]. Team culture, as the context that configures team functioning through its influence on team members, is expected to impact on learning behaviours such that different cultures influence team learning behaviours in a different way. 

However, most of the current studies that have focused on this issue are cross-sectional [19,20] and do not provide any information about changes in learning behaviours over time. Because teams are embedded within a temporal context (e.g., a deadline to carry out a task in project teams), time is a fundamental component of teams’ relevant context [23,24,25,26]. Team learning is a dynamic and temporal set of team behaviours that can be influenced by different variables at multiple levels (organizational level, team level, or individual level), generating changes or improvements over time for the team, its members, and the organization [14]. These behaviours emerge and evolve through team members’ interactions [3]; however, there are relatively few empirical studies on behavioural change in teams [24].

The purpose of this study is two-fold: first, following the literature on team development, e.g., [27,28] and temporal process phases [29], which suggest that all teams change over time and that different processes are activated at different times based on the specific demands of the team’s tasks, this study is set up to enhance knowledge about the use of learning behaviours over time, as very few studies have analysed the temporal effect; second, following social constructivism, which suggests that the context shapes what is learnt and how it is learnt and what is regarded as important, our study aims to explore the influence of team culture on the use of learning behaviours within project teams. Specifically, we pursue the following research questions: (a) to what extent do teams use learning behaviours over time? and (b) how does team culture affect team learning behaviours?

Knowing the extent to which teams use learning behaviours over time could offer researchers and practitioners information to create facilitation interventions to improve the use of team learning behaviours within teams. 

### 1.1. Team Learning

Team learning could be conceived as encompassing two sides: one concerns learning behaviours that occur in members’ interaction, through which they collectively identify, discuss, and solve problems to provide solutions [30]; the other concerns the outcomes that emerge as a collective property of the team, such as the team’s shared cognitions, which are built through members’ participation in team learning activities [31]. 

This study is focused on the former, considering team learning as a process that occurs through members’ interactions and behaviours. According to [32] (p. 244), “there is general agreement that the exchange of information between team members is a key activity” of the team learning process; however, a lack of consensus exists around which other behaviours are involved in this process. Nevertheless, one of the most quoted conceptualizations is from [10], who describes team learning as an ongoing process of collective reflection and action characterized by several team behaviours, namely, exploring, reflecting, discussing errors and unexpected outcomes of actions, seeking feedback, and experimenting within and as a team. Therefore, any conceptualization of team learning should include the ability of team members themselves to acquire knowledge and competencies, as well as their ability to share that information with the rest of the team [9]. It should be noted that learning involves interaction amongst team members as well as with others external to the team and with the environment [7]. 

Furthermore, research has shown that teams can differ considerably in the extent to which they engage in learning behaviours [10,33], highlighting the relevance of studying the factors that can explain this variance between teams and even in the same team over time. Because organizations are currently regarded more as being composed of teams rather than individuals [34], the culture of the team should be considered as a factor that can explain team learning behaviours. Team culture emerges from the underlying assumptions and beliefs of team members about what they share, and different types of culture can influence in distinct ways how team members perceive their work and roles [35]. In this sense, team culture shapes the attitudes and, consequently, the behaviours of team members, becoming an important factor in understanding learning behaviours.

### 1.2. Team Learning and Team Culture

Previous studies have shown that specific elements of organizational culture, are related, directly and indirectly, to team learning ([35] for a review). Considering this and given that any group of workers can develop their own culture [36] we can consider the culture of the teams as a condition to team learning [20], since it is related to what drives team learning [37]. Team culture is an emergent and simplified set of rules, values and actions, work capability expectations, and member perceptions that individuals within a team develop, share, and enact after mutual interactions [38]. It prescribes the way in which work is approached in the team and is related to work attitudes, perceptions, and behaviours [39], playing a crucial role in shaping teamwork. 

Previous research has shown that team culture has a significant relationship with team functioning and team outcomes (e.g., [40,41]); however, little is known about the link between culture and learning at the team level [19,20], despite the fact that this relationship has already been the subject of several studies at an organizational level. Research findings highlight that some cultural characteristics can be facilitators of or hindrances to learning [42,43,44]. For example, clan, adhocracy, and market cultures, which are different culture orientations, i.e., each orientation corresponds to a view of culture as a set of assumptions and deep-level values, have a positive effect on organizational learning, whereas hierarchy culture does not show any impact on it [45]. Moreover, the same cultural aspect can facilitate a type of learning while hindering another, as shown, for instance, by [46], in a study of 56 business development teams. The author found that learning was more effective in teams operating with autonomy with respect to goals and supervision. 

As with organizational culture, team culture can be categorized into different types or orientations [47] playing unique, independent roles in predicting different team outcomes [40]. Recently, ref. [48] found that team cultural orientations are positively related to learning behaviours, with cultures oriented to innovation and goals assuming a prominent role in promoting learning in teams. Conceiving culture as an attribute that distinguishes groups and organizations from one another [49] provides the opportunity to examine how culture impacts and predicts team learning behaviours. Based on the literature that suggests that team culture orientations impact team processes and outcomes in a team differently, it is our aim to determine if different aspects of team culture have a different effect on team learning behaviours over time.

### 1.3. Team Learning over Time

Past research has suggested that learning behaviours may occur differently depending on where the team stands in its lifecycle [14]; moreover, as we have already mentioned, teams can differ considerably in the extent to which they (intentionally or not) engage in learning behaviours [10,13,33]. In fact, as complex, adaptive, and dynamic systems, teams are interdependent entities embedded in a hierarchy of levels revealing complex behaviours, continuously adapting to environmental changes, and whose functioning is dependent both on the team’s history and on its anticipated future [50]. In other words, teams change over time, and, in this sense, team learning can be better understood as a dynamic process that can be attributed to the team as an entity, changing as the team changes [51]. For example, ref. [52] examined team learning curves, which imply the progressive development of team learning over time. Previously, [11] noted that team members can perceive time as an individual resource or as a dimension of learning belonging to the team. Although not explicitly measuring team learning, these studies imply that teams may differ in the use of learning behaviours over time. 

Teams, as a whole, change systematically over time [23]; moreover, based on the theoretical contributions of [11,52], we expect a change in the use of team learning behaviours over time. In other words, to better understand the importance of learning behaviours in a team context, it is important to understand changes in these processes from a temporal perspective, creating a more in-depth understanding of team learning behaviours throughout the lifespan of team projects. 

Considering this, an analysis of the role of time in team learning behaviours is required [26,53,54]. However, only a few studies have studied how team learning behaviours change over time [55,56]. Given this, the present study is set up to analyse how teams differ in the use of learning behaviours over time and if team culture orientations have an effect.

## 2. Materials and Methods

### 2.1. Participants and Data Collection

Data were collected from 33 university project teams (comprising a total of 154 members), structured according to the Project Based Learning (PBL) model, made up of undergraduate students of engineering and technology courses from one Portuguese higher education institution. Project teams were chosen because they are teams that are composed for the purpose of completing a one-time goal, whose members only have a short amount of time to become familiar with each other [1,57]. Regarding PBL, it refers to an inquiry-based instructional method that engages learners in knowledge construction by having them accomplish meaningful projects and develop real-world products [58]. In other words, it is a teaching method in which complex real-world problems are used as a vehicle to promote student learning. At the university where data was collected, projects are developed in small groups of students, which are assigned a small physical space to meet and work. Students are also granted extended access to laboratories, evenings and weekends included. Each group has a supervisor for every project carried out. The supervisor’s role is to facilitate students’ progression, to guide without disclosing the solution, to help by asking meaningful directing questions. These kinds of teams are suitable for studying team learning because students are presented with challenges that have no a priori solution. This means that they need to explore, set goals, collect resources, try different solutions, share knowledge, fail, and proceed until an acceptable solution is found and implemented. Thus, students need to learn together throughout the development of the project. Indeed, the PBL method is designed to promote problem-solving and collective learning [59].

Teams were composed of 3 up to 6 members (M = 3; SD = 1.4), with 67% from electrotechnical engineering and 33% from the information technology degree; 25.7% were in the first year, 17.4% in the second year, and 56.9% in the third year of their program. The mean age was 23.9 years (SD = 5.9), and 88.2 % of the team members were male and the majority were full time students (79.9%). 

Data were collected at three moments: at the beginning (T0), in the middle (T1), and at the end (T2) of the academic semester, corresponding to the timeline of the work team. In a meeting with each team, team members answered a questionnaire together by reaching a consensus. Participation in the study was voluntary. All participants provided their informed consent; confidentiality and anonymity were guaranteed by the research team. 

### 2.2. Measures

Since this study implied collecting data longitudinally, we chose to use Visual Analogue Scales (VASs). Following the criteria suggested by [60], for each of the constructs, or, if it was the case, for each dimension of the construct, the original multi-item measures on which we based our questionnaires, were transformed into a single-item measure [61]. Our option was in line with the suggestion of [62], who suggested that multi-item questionnaires may be less suitable for capturing change in teams over time because they are made to reliably differentiate between people and not to detect change. The referred authors even stated that “the more items are used, the more these instruments are vulnerable to lack of measurement equivalence over time. An alternative is to develop single-item measures, e.g., using graphic scales” (p. 643). However, since we are aware of some of the psychometric shortcomings of single-item measures, content and face validities were analysed through three pilot studies. No problems were identified [63]. Convergent validity studies with the correspondent multi-item measures [60] and nomological validity studies were also carried out [64,65].

Hence, to measure team culture orientations, four single-item measures based on the FOCUS questionnaire [66], an international questionnaire for measuring organizational culture based on a [67] competing values model, were developed. Each item corresponds to a team culture orientation: support orientation, characterized by values such as cooperation, team spirit, participation; goal orientation, which emphasizes aspects such as performance indicators, accomplishment, rationality; rules orientation, characterized by respect for authority, division of work; innovation orientation, where creativity, openness to change, and anticipation are valued. By reaching a consensus, team members had to mark the presence of each one of the cultural orientations on a VAS from 0 to 10 at the three data collection moments. All the items were formulated using the terms “we” and not “I”. We were aware of some disadvantages of the group consensus, such as the problem of social desirability and conformity. However, in order to decrease the probability of phenomena such as social desirability and conformity, some procedures suggested by [68] were applied by the research team: in the introduction, each one of the team members was encouraged to (a) answer individually before the group discussion; (b) speak their minds and not worry about others’ opinions; (c) agree with others only when comfortable with the decision about the answer chosen; (d) discuss and explore the answers proposed by each one of the group members without a sense of urgency. Additionally, after the completion of the questionnaire by the group, the research team questioned the group regarding the decision process and if, indeed, all group members agreed with the questionnaire answers. The sample used for the validity studies was comprised of 250 Portuguese higher education students. Regarding convergent validity, the correlations between the single-item measure and the respective multi-item dimension of the FOCUS questionnaire ranged from 0.53 to 0.67 (support = 0.67, innovation = 0.53, rules = 0.53 and goal = 0.59). Concerning nomological validity, the single-item measures for support and innovation orientations correlate with affective commitment (r = 0.61 and r = 0.49, respectively), as expected, e.g., [69,70].

Team learning was measured using five single items adapted from the multi-item Team Learning Behaviours’ Instrument [12] which, in turn, was based on [10] five types of learning behaviours, complemented with the contributions of other authors [31,70,71,72]. Thus, the five single-item measures correspond to the following learning behaviours: exploring, conceived as conversational actions to share knowledge and opinions and constructively managing opinion differences; feedback, characterized as seeking and analysing feedback, internally and externally, to measure whether the team is working well and drawing conclusions that lead to further exploration and experimentation; collective reflection, seen as collectively looking back on or ahead to experiences, goals, actions, working methods, strategies and assumptions to discuss; error management, conceived as discussing errors collectively and exploring how to prevent them; experimenting, considered as collectively doing things differently from before. Concerning convergent validity, a sample of 212 Portuguese higher education students was used. The correlations between the single-item measure and the respective multi-item dimension of the Team Learning Behaviours’ Instrument ranged from 0.48 to 0.68 (exploring = 0.64, collective reflection = 0.54, error management = 0.68, feedback = 0.48 and experimenting = 0.49), offering satisfactory convergent validity. Concerning nomological validity, the five single-item measures correlate with team satisfaction, ranging from 0.50 to 0.72, as expected from previous research with team learning multi-item instruments, e.g., [73,74]. These results offer satisfactory confidence in the use of these single-item measures.

### 2.3. Analysis

Data analysis was conducted with R software [75] through growth modelling. Behind growth models is the idea that individuals (in this case, teams) differ over time [76]. Following the recommendations of [77], level-1 analyses were conducted at the intragroup level. The predictor modelled in the analyses was time. The objective was to examine how the dependent variable (DV) (in this case, team learning behaviours) changes over time. In other words, to ascertain if there was a pattern in the relationship between time and team learning behaviours in the teams of the sample. Level-2 analyses were conducted at the intergroup level, and the predictors included were time and culture orientations. The objective was to examine if the independent variable (IV) (in this case, team culture) could help to explain the extent of the use of team learning behaviours over time [77]. Culture is a relatively stable aspect of team life due to the difficulty of changing the shared taken-for-granted assumptions created by the team. When a hypothesis or a trial to solve something is implemented, and that implementation repeatedly shows positive results (i.e., is well succeeded) [74], it gradually becomes a guide for action and a “way of doing things” in the team. Thus, considering the expected stability of team culture over time, our option was to include the measurement of team cultural orientations only at T0 (which roughly captures the first month of team functioning) in this step of the analysis.

## 3. Results

Table 1 displays the means, standard deviations, and inter-correlations for study variables. In general, a high level of engagement in team learning behaviours over time was found. We found significant correlations between team learning behaviours and culture orientations only at T0. Goals and support orientations correlate significantly with all team learning behaviours and innovation correlate significantly with exploring, collective reflective, error management, and experimenting. Rules orientation did not display significant correlations with team learning behaviours. 

In the analyses aimed at modelling the relationship between time and learning behaviours, we only found a significant linear relationship between time and the experimenting behaviour (*b* = 0.75, *t* (64) = 2.50, *p* = 0.015), such that experimenting increases over time. Concerning the other four learning behaviours, no significant relationships with time were found, that is to say, a pattern in these behaviours was not detected over time in the sample teams (e.g., a pattern of growth, a pattern of decrease, or a non-linear pattern).

In the analyses regarding the interaction effect of time and cultural orientations on learning behaviours, we only analysed experimenting as DV, considering that it was the only learning behaviour which showed a statistically significant pattern over time. We found a significant interaction effect of time and support orientation, *b* = −0.35, *t* (61) = −2.85, *p* = 0.006; of time and innovation orientation, *b* = −0.38, *t* (61) = −2.49, *p* = 0.016; and of time and goal orientation interaction, *b* = −0.35, *t* (55) = −2.64, *p* = 0.001. No significant interaction effect of time and rules orientation was found (Table 2). 

Team culture orientations were measured in a scale of 10-centimenter, which was then divided in five levels for analysis purposes (levels 1, 3, 5, 7, 9). Figure 1, Figure 2, Figure 3 and Figure 4 show the interaction plots between time and cultural orientations. In each graph, we can see the five levels considered in the scale of each cultural orientation. In a similar way, teams more oriented to support and innovation at T0 make more use of experimenting and tend to show a balance over time, i.e., tend to show high levels of experimenting learning behaviours throughout the time of carrying out the project.

On the other side, teams characterized by cultures less oriented to support and innovation at T0 make less use of experimenting but showed the strongest increase over time. That is, the closer teams are to the deadline, the more they use experimenting behaviours.

Similarly, teams more oriented to goals at T0 make more use of experimenting and tended to be constant over time. Cultures less oriented to goals at T0 report less initial use of experimenting but showed a more pronounced increase over time. In other words, the less teams are oriented to goals at T0, the greater is the likelihood of increasing experimenting over time. It should be noted that, independent of being characterized by high, medium, or low levels of goals orientation at T0, all teams tend to show the same level of experimenting behaviours at the end of the project. As already mentioned, no significant interaction effect of time and rules orientation was found. In other words, despite starting at different degrees, the relationship between rules and experimenting over time is the same at all different levels of the variable rules. 

All in all, the graphs show that these three cultural orientations (support, innovation, and goals), if developed when the teams start functioning, lead to a constant use of experimenting over time, that is, a high use of experimenting behaviours early, at the halfway point, and also near the end of the teamwork. In contrast, teams that do not develop these cultural orientations when they start to work show lower use of experimenting behaviours at the beginning. However, these teams show a higher increase over time in experimenting behaviours, reaching or even slightly exceeding the levels of the teams which were initially high in these cultural orientations.

## 4. Discussion

The purpose of this article was to examine team learning behaviours in project teams, focusing on the extent to which the use of these behaviours changes over time, as well as on the influence of team culture on the use that teams make of learning behaviours. The results only show a linear relationship between time and experimenting, whereas the use of experimenting tends to increase over time. No significant relationship was found between time and the other four team learning behaviours. As suggested by [78], the efficiency required to meet deadlines and achieve high performance in project teams might lead team members to give priority to achieve performance goals. Because our sample was composed of project teams that have a very short amount of time to complete the task, team members had to foster experimenting through time, and not so much at the beginning of the project. 

Furthermore, we identified that support, innovation, and goal emerged as being expressly related to experimenting, such that teams less oriented to support, innovation, and goals at the beginning of their functioning make less use of experimenting but showed a more pronounced increase over time. Teams more oriented to support, innovation, and goals when they are just starting to work together make more use of experimenting, showing a constant use of this behaviour over time. Accordingly, considering this result and the characteristics of support culture, innovation culture and goals culture, our findings suggest that the building of mutual understanding and support, good interpersonal relationships, creativity, openness to change, and the pursuit of alternative ways to perform tasks and solve problems, as well as an orientation to tasks and well-defined objectives at the beginning of the workgroup, are conditions to experiment with or reflect on working methods, not only from the beginning, but also over time. In a setting such as project teams, where creativity, exploration, and experimentation are critical, support, innovation, and orientation to goals appear to be effective right from the beginning, motivating team members to engage in experimenting activities. Teams characterized by those three cultural orientations are the teams that also show experimenting behaviours from the beginning and that maintain those types of behaviours over time. These results are in accordance with those of other studies. For example, ref. [45] concluded that, to increase learning and innovativeness, organizations should focus on building a culture that incorporates teamwork, a sense of competitiveness, employee involvement programmes, and risk taking. Thus, encouraging team members to cooperate and support each other, developing openness to change and innovative strategies, and also being focused on task accomplishment and performance indicators constitute a good strategy to increase teams’ ability to experiment with new ways of doing things. More recently, ref. [48] found that support, innovation, rules, and goal orientations are positively related to team learning, with innovation and goals orientation seeming to be more crucial in promoting learning in teams. 

On the other side, teams less oriented to support, innovation, and goals at the beginning of the work only equate or even slightly surpass other teams at the end of the project. This result might be explained by the task definition and temporary nature of teamwork. In fact, based on the approach proposed by [27], we can hypothesize that the members’ awareness of time and deadlines boost their experimenting behaviours. According to this model, the passage of time influences team functioning, particularly in temporary teams, especially around mid-point transitions, where team members experience temporal urgency [27]. In other words, the notion of time can form an important causal factor because, according to the model, the group’s progress occurs based on the awareness of its members regarding the past and future time, as well as on a consideration of the deadlines established. Therefore, teams are likely to adapt their strategies based on their perceptions of anticipated time horizons Moreover, ref. [79] found that teams displayed significantly more learning behaviours later, rather than earlier, in their work because of the time it took the team to develop social conditions such as psychological safety and potency. 

Rules orientation was the only variable not related to experimenting behaviours. Ref. [45] found that hierarchy culture, characterized by formalized rules and policies and a structured workplace wherein employees do not have autonomy to perform their job, does not show any impact on learning. Probably, in team cultures characterized by compliance with established standards/rules, well-defined working procedures, and respect for guidelines, where team members are obliged to follow conventional methods to address the issues, the formalized and centralized structure with standard action procedures does not allow them to approach things from different and new perspectives, and, therefore, does not provide the opportunity to learn new things [45]. 

## 5. Conclusions

The implications of this study for future research and practice are important. On one hand, the analysis of team learning over time, scarcely covered in the literature, has shown the importance that time has for team learning, particularly, for experimenting behaviours. On the other hand, focusing on team-level constructs, this study surpasses the limit of previous studies that are focused on the individual and the organization. That is, one of our goals was to analyse learning and culture, considering both constructs from the team perspective analysis, which has rarely been performed in other studies. By examining how teams learn over time, this research answers calls for a better understanding of team learning as a dynamic construct [53]. Mainly, results show that time and team culture are important determinants of team learning behaviours, particularly experimenting behaviours. The findings show that team learning does not operate uniformly over the lifetime of project teams, depending on the culture of the team. This has important implications for the current literature of team learning and knowledge work research, which has tended to rely on experimental and non-longitudinal research designs using single-point measures of team learning. In contrast to using a single snapshot measure of team learning, more research is needed that examines team learning as an ongoing temporal phenomenon that fluctuates throughout the project. 

Nevertheless, some remarks can be made concerning the present study. Following [80], we consider that student samples can provide valuable information and enhance tests of the theory. For example, ref. [81]’s review revealed that many of the results across student teams parallel the findings of other team types concerning team design characteristics and team performance. Similarly, concerning team learning behaviours, recently, ref. [82] compared student vs. non-student samples, concluding that its analysis is equally valid across different sample types. Nonetheless, because every team type has specific characteristics, future research is necessary to replicate this study with different team types, to compare the findings and test if team culture acts similarly on team learning behaviours in other teams as in students’ project teams. Moreover, our choice of variables to include in the present study has led to the exclusion of other variables. Multiple inputs such as team psychological safety, interdependence, or team potency may affect team learning behaviours [10,31]. The present study is founded on perceptions of the team members. It is recommended to measure the constructs by questioning different stakeholders of the team (internal or external) or even to use different methods of data gathering, such as observation. Finally, considering the small number of teams in our sample, we used separate models for the different team culture orientations. Although the Competing Values Framework—the theoretical model that comprises the four cultural orientations in use—distinguishes them as unique sets of behaviours, values, beliefs, and assumptions that are expected to have different effects on the outcome [83], further research using larger samples to examine the robustness of our findings is required.

The emphasis of this study was on behaviours and, in this sense, from a practical perspective, the results obtained should be useful for orienting teams and team leaders about the kind of behaviours they should exhibit and the values they should develop and share to learn and improve their functioning over time. Establishing a clear shared understanding of team functioning and creating a hybrid culture can be a good strategy. 

Additionally, this study was set up to enhance knowledge about the changes in the occurrence of team learning behaviours over time, extending research in teamwork and team learning. Particularly, it highlights the need to be aware of the dynamic nature of team learning and the importance of team culture. Team leaders need to assess and understand what aspects of team culture at the beginning of the team need to be considered when developing learning initiatives.

This study’s results seem to suggest that there is a significant linear relationship between time and experimenting, and that team culture might explain some of the variation in the time–experimenting slope. Considering our findings, team cultures that promote mutual understanding and good interpersonal relationships, goal setting and the accomplishment of objectives, flexibility, creativity, and the search for alternative ways to perform tasks/problem solving are more conducive to developing learning behaviours of experimenting right from the beginning of the team.

## Figures and Tables

**Figure 1 behavsci-12-00449-f001:**
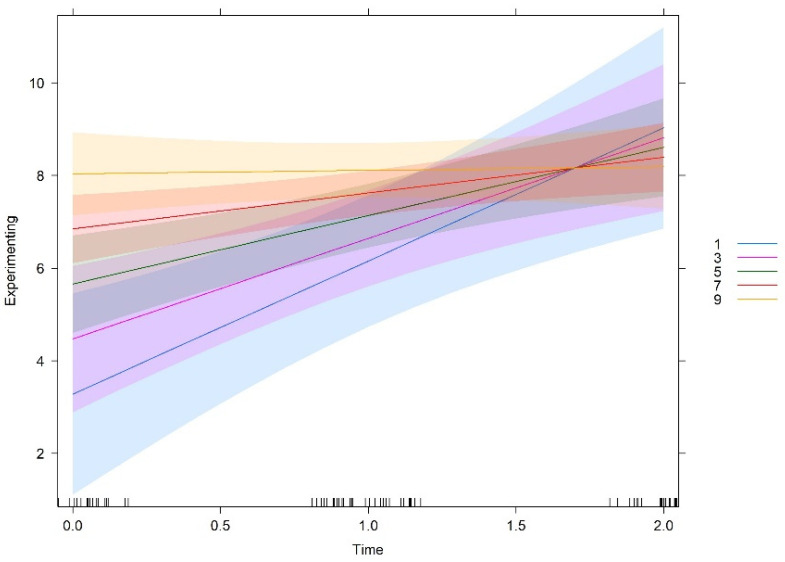
Interaction effect between time and support.

**Figure 2 behavsci-12-00449-f002:**
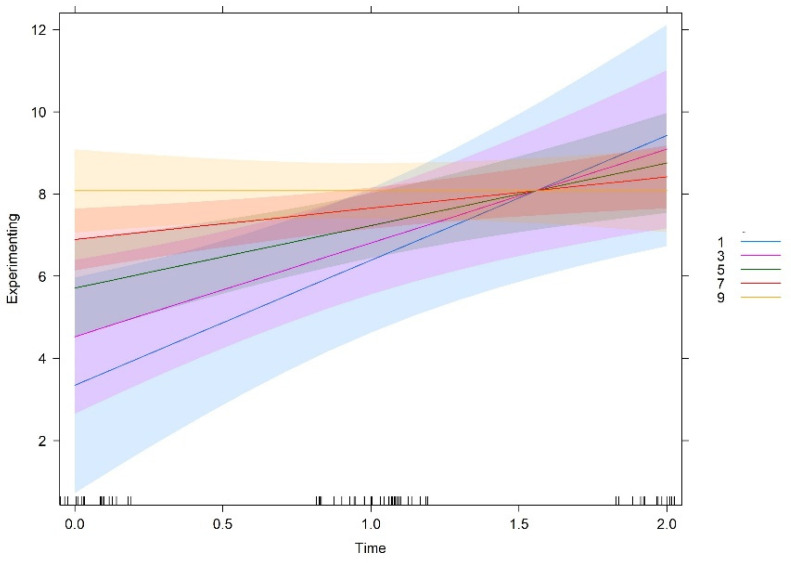
Interaction effect between time and innovation.

**Figure 3 behavsci-12-00449-f003:**
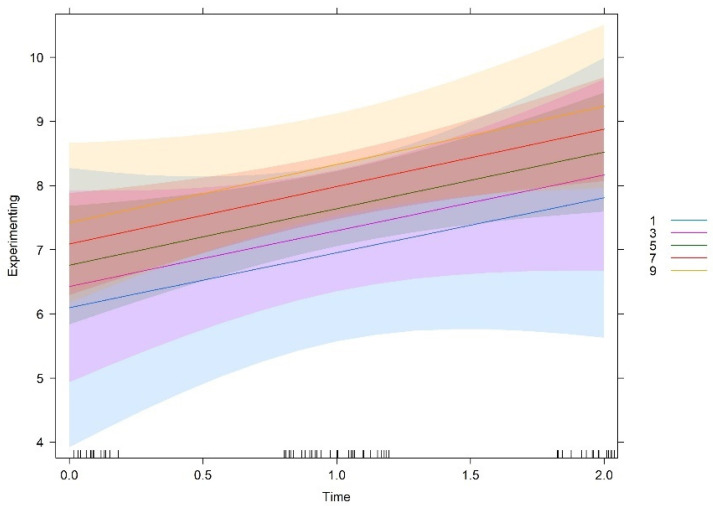
Interaction effect between time and rules.

**Figure 4 behavsci-12-00449-f004:**
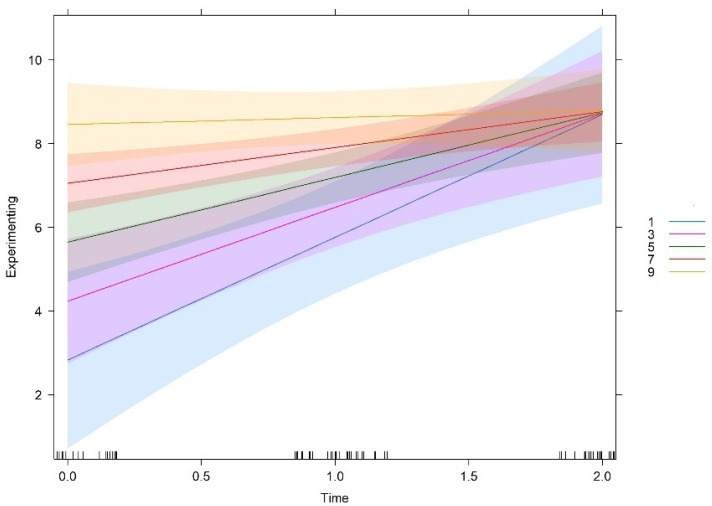
Interaction effect between time and goal.

**Table 1 behavsci-12-00449-t001:** Means, standard deviations, and correlations between study variables.

	M	SD	1	2	3	4	5	6	7	8	9	10	11	12	13	14	15	16	17	18	19
**1. Exploring_T0**	8.69	1.79	-																		
**2. CR_T0**	8.24	1.90	0.77 **	-																	
**3. EM_T0**	7.88	2.20	0.84 **	0.68 **	-																
**4. Feedback_T0**	8.01	1.65	0.49 **	0.48 **	0.47 **	-															
**5. Experimenting_T0**	6.58	2.80	0.38 *	0.21	0.48 **	0.51 **	-														
**6. Exploring_T1**	8.76	1.31	0.32	−0.02	0.38 *	0.16	0.41 *	-													
**7. CR_T1**	8.30	1.76	0.09	−0.15	0.19	0.09	0.19	0.67 **	-												
**8. EM_T1**	8.14	1.92	0.15	0.03	0.23	0.06	0.33	0.61 **	0.50 **	-											
**9. Feedback_T1**	8.42	1.81	0.103	0.05	0.07	0.12	0.24	0.44 *	0.27	0.47 **	-										
**10. Experimenting_T1**	8.26	1.98	0.18	0.15	0.10	0.13	0.17	0.55 **	0.39 *	0.52 **	0.45 **	-									
**11. Exploring_T2**	8.99	1.05	0.25	0.29	0.15	0.29	0.31	0.17	0.03	−0.01	0.27	0.17	-								
**12. CR_T2**	8.56	1.22	0.22	0.34	0.14	0.18	0.24	0.15	0.06	0.13	0.44 *	0.42 *	0.81 **	-							
**13. EM_T2**	8.41	1.38	−0.02	0.10	−0.11	0.04	0.05	0.19	−0.03	0.26	0.34	0.50 **	0.29	0.53 **	-						
**14. Feedback_T2**	8.28	1.80	0.00	−0.03	−0.05	0.22	0.12	0.26	0.07	0.42 *	0.51 **	0.45 **	0.41 *	0.49 **	0.56 **	-					
**15. Experimenting_T2**	8.07	2.42	0.18	0.17	−0.05	0.13	−0.15	0.07	0.06	0.06	0.20	0.23	0.11	0.06	0.50 **	0.22	-				
**16. Support_T0**	7.34	2.29	0.74 **	0.59 **	0.76 **	0.49 **	0.56 **	0.30	0.01	0.09	0.13	0.18	0.22	0.28	−0.00	−0.10	−0.06	-			
**17. Innovation_T0**	7.28	1.89	0.46 **	0.50 **	0.47 **	0.21	0.52 **	0.11	−0.12	0.15	0.18	−0.03	0.16	0.11	−0.13	−0.14	−0.03	0.47 **	-		
**18. Rules_T0**	6.40	2.04	0.14	0.13	0.11	0.14	0.24	−0.22	−0.16	−0.10	0.04	−0.13	-0.27	−0.21	0.05	−0.26	0.32	0.19	0.20	-	
**19. Goal_T0**	6.92	2.10	0.59 **	0.53 **	0.70 **	0.46 *	0.63 **	0.16	−0.01	0.16	0.21	0.17	0.02	0.16	0.15	0.08	0.12	0.72 **	0.52 **	0.54 **	-

Note. * *p* < 0.05; ** *p* < 0.01, N = 33 teams; measures’ range: from 0 to 10. CR. Collective Reflexive; EM. Error Management.

**Table 2 behavsci-12-00449-t002:** Multilevel model for the interaction effect Time x Team culture orientations (support, innovation, goal, and rules) on experimenting.

	B	SE b	df	*t* Value	*p*
Initial Level, β0	2.68	1.25	61	2.15	0.036
Time, β1	3.22	0.95	61	3.41	0.001
Support, β2	0.59	0.16	30	3.67	0.001
Time × Support, β3	−0.35	0.12	61	−2.85	0.006
Initial Level, β_0_	2.75	1.52	61	1.82	0.074
Time, β_1_	3.42	1.15	61	2.97	0.004
Innovation, β_2_	0.59	0.20	30	2.94	0.006
Time × Innovation, β_3_	−0.038	0.15	61	−2.49	0.016
Initial Level, β_0_	2.13	1.22	55	4.66	0.087
Time, β_1_	3.28	0.95	55	0.86	0.001
Goal, β_2_	0.70	0.17	27	0.87	0.000
Time × Goal, β_3_	−0.35	0.13	55	−2.64	0.001
Initial Level, β_0_	5.93	1.27	55	1.74	0.000
Time, β_1_	0.85	0.99	55	3.46	0.394
Rules, β_2_	0.17	0.19	27	4.15	0.392
Time × Rules, β_3_	001	0.15	55	0.04	0.966

Note. N = 33 teams.

## Data Availability

Not applicable.

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
