# Peer review of "Dynamics of Team Learning Behaviours: The Effect of Time and Team Culture"

_behavsci, 2022, doi:10.3390/bs12110449_

Round 1
Reviewer 1 Report
It is not clear how are the Vais (2014) and Melo (2015) Master theses are related to this work. Also they are not in references and it should not be an APA-style quote. Is the footnote at the end a leftover from another article?
Reviewer 2 Report
Thank you for the opportunity to review your manuscript on team learning behavior as function of team culture and time, which shows very interesting findings. Please find below some suggestions.
First, the theoretical positioning of your study is very brief (p.2 second paragraph). The theoretical positioning is rather brief (top p2). You argue that time is a fundamental component of team's context, and that all teams change over time. However, little is mentioned about what literature you exactly contribute to, what we do know and do not know about your approach, and what we would miss without your contribution. Also the conclusions and implications section is very brief on this: what lessons can be drawn from your study?
Second, there are several methodological concerns. Little information is provided about the group formation, while it comes with many concerns: If the student groups are part of the same program and perhaps even collaborated before, their learning behavior most likely started before T0. On the other hand, if the students met for the first time at T0, it seems unlikely that they already formed a team culture, implying that there are serious concerns about at least one of the independent variables. Next, the paper lacks discussion on how the exact team task and the student environment relate to the learning behavior. It also remains unclear how individual scores are aggregated to team scores. In terms of modeling, the study uses separate models for different team culture items, while these items are strongly correlated with each other (i.e. risk of spurious effects). In addition, the study examines many potential effects without applying Bonferroni adjustments, although the small sample size might complicate significance testing anyhow. The graphs used to discuss the interaction effects are much appreciated.
Finally some minor issues regarding the writing. Particularly the first sentence is remarkably long. Further, there is some inconsistency in citation, with authors mentioned in some case, "According to Cohen and Baily [1]" (p1) and not mentioned in an identical other case "According to [31]" (p2).
Reviewer 3 Report
Thank you for the opportunity to review this manuscript entitled Dynamics of team learning behaviours: the effect of time and 2 team culture".
Your topic is very relevant for Behavioral Sciences. I believe that your work is well thought out, supported by relevant literature and well structured. I only have two questions:
(1) How does your research relate to the life cycle of the team, which also shows different behavior of teams depending on the phase they are in?
(2) The teams you analyzed worked in accordance with the problem based learning. Please explain the difference between problem based leraning and project based learning. Did the students included in the analyzed projects come from the same fields of study and specializations?
Round 2
Reviewer 2 Report
The study has much improved and makes an interesting contribution that is adequately discussed.
The sample size remains a concern, but is adequately discussed as limitation. Hence, all my prior concerns are addressed.
For future research, it might be interesting to explore non-linear (e.g. curvilinear or punctuated) relationships between time and the remaining four learning behaviors, but that is beyond the scope of this paper.